# Towards certifying $\ell_\infty$ robustness using Neural networks with $\ell_\infty$-dist Neurons

## Abstract

It is well-known that standard neural networks, even with a high classification accuracy, are vulnerable to small $\ell_\infty$ perturbations. Many attempts have been tried to learn a network that can resist such adversarial attacks. However, most previous works either can only provide empirical verification of the defense to a particular attack method, or can only develop a theoretical guarantee of the model robustness in limited scenarios. In this paper, we develop a theoretically principled neural network that inherently resists $\ell_\infty$ perturbations. In particular, we design a novel neuron that uses $\ell_\infty$ distance as its basic operation, which we call $\ell_\infty$-dist neuron. We show that the $\ell_\infty$-dist neuron is naturally a 1-Lipschitz function with respect to the $\ell_\infty$ norm, and the neural networks constructed with $\ell_\infty$-dist neuron ($\ell_\infty$-dist Nets) enjoy the same property. This directly provides a theoretical guarantee of the certified robustness based on the margin of the prediction outputs. We further prove that the $\ell_\infty$-dist Nets have enough expressiveness power to approximate any 1-Lipschitz function, and can generalize well as the robust test error can be upper-bounded by the performance of a large margin classifier on the training data. Preliminary experiments show that even without the help of adversarial training, the learned networks with high classification accuracy are already provably robust.

## 1 Introduction

Modern neural networks are usually sensitive to small, adversarially chosen perturbations to the inputs (Szegedy et al., 2013; Biggio et al., 2013). Given an image $x$ that is correctly classified by a neural network, a malicious attacker may find a small adversarial perturbation $\delta$ such that the perturbed image $x + \delta$, though visually indistinguishable from the original image, is assigned to a wrong class with high confidence by the network. Such vulnerability creates security concerns in many real-world applications.

Developing a model that can resist small $\ell_\infty$ perturbations has been extensively studied in the literature. Adversarial training methods (Szegedy et al., 2013; Madry et al., 2017; Goodfellow et al., 2015; Huang et al., 2015; Athalye et al., 2018; Ding et al., 2020) first learn on-the-fly adversarial examples of the inputs, and then update model parameters using these perturbed samples together with the original labels. Such approaches are restricted to a particular (class of) attack method and cannot be formally guaranteed whether the resulting model is robust against other attacks. Another line of algorithms trains robust models by maximizing the certified radius provided by robust certification methods. Weng et al. (2018); Wong & Kolter (2018a); Zhang et al. (2018); Mirman et al. (2018); Wang et al. (2018); Gowal et al. (2018); Zhang et al. (2019b) develop their methods based on linear or convex relaxations of fully connected ReLU networks. However, the certification methods are usually computationally expensive and can only handle ReLU activations. Cohen et al. (2019); Salman et al. (2019); Zhai et al. (2020) show that a certified guarantee on small $\ell_2$ perturbations can be easily computed for general Gaussian smoothed classifiers. But recent works suggest that such methods are hard to extend to the $\ell_\infty$-perturbation scenario.

In this work, we overcome the challenge mentioned above by introducing a new type of neural network that naturally resists local adversarial attacks and can be easily certified under the $\ell_\infty$ perturbation. In particular, we propose a novel neuron called $\ell_\infty$-dist neuron. Unlike the standard neuron design that uses a non-linear activation after a linear transformation, the $\ell_\infty$-dist neuron is purely based on computing the $\ell_\infty$ distance between the inputs and the parameters. It is straightfor-

ward to see that such a neuron is 1-Lipschitz with respect to the $\ell_\infty$ norm and the neural networks constructed with $\ell_\infty$-dist neuron ($\ell_\infty$-dist Nets) enjoy the same property. Based on such a property, we can obtain the certified robustness for any $\ell_\infty$-dist Nets using the margin of the prediction outputs.

Theoretically, we investigate the expressive power of $\ell_\infty$-dist Nets and its adversarially robust generalization ability. We first prove a Lipschitz-universal approximation theorem for $\ell_\infty$-dist Net using a structured approach, which shows that $\ell_\infty$-dist Nets can approximate any 1-Lipschitz function with respect to $\ell_\infty$ norm. We then give upper bounds of robust test error based on the Rademacher complexity, which shows that the robust test error would be small if the $\ell_\infty$-dist Net learns a large margin classifier on the training data. Both results demonstrate the excellent capability and generalization ability of the $\ell_\infty$-dist Net function class.

The $\ell_\infty$-dist Nets have nice theoretical guarantees, but empirically, training an $\ell_\infty$-dist Net is not easy. For example, the gradient of the parameters in the $\ell_\infty$ norm is sparse, which makes the optimization inefficient. We show how to initialize the model parameters, apply proper normalization, and overcome the sparse gradient problem via smoothed approximated gradients. Preliminary experiments on MNIST and Fashion-MNIST show that even without using adversarial training, the learned networks are already provably robust.

Our contributions are summarized as follows:

- We propose a novel neural network using $\ell_\infty$-dist neurons, called $\ell_\infty$-dist Nets. Theoretically,

  - In Section 3, we show that $\ell_\infty$-dist Nets are 1-Lipschitz with respect to the $\ell_\infty$ norm in nature, which directly guarantees the certified robustness of any $\ell_\infty$-dist Net (with respect to the $\ell_\infty$ norm).
  - In Section 4.1, we prove that $\ell_\infty$-dist Nets have good expressive power as it can approximate any 1-Lipschitz function with respect to the $\ell_\infty$ norm.
  - In Section 4.2, we prove that $\ell_\infty$-dist Nets have good generalization ability as the robust test error can be upper-bounded by the performance of a large margin classifier on the training data.

- We provide several implementation strategies which are shown to be practically helpful for model training (in Section 5).

## 2 RELATED WORKS

There are two major lines of works seeking to get robust neural networks:

**Robust Training Approaches.** Previous works showed that the conventional neural networks learned using standard training approaches (e.g., maximum likelihood method) are sensitive to small adversarial perturbations, and significant efforts have been put on developing training approaches for learning robust models. Adversarial training is the most successful method against adversarial attacks. By adding adversarial examples to the training set on the fly, adversarial training methods (Szegedy et al., 2013; Goodfellow et al., 2015; Huang et al., 2015; Zhang et al., 2019a; Wong et al., 2020) can significantly improve the robustness of the conventional neural networks. However, all the methods above are evaluated according to the empirical robust accuracy against pre-defined adversarial attack algorithms, such as projected gradient decent. It cannot be formally guaranteed whether the resulting model is also robust against other attacks.

**Certified Robustness.** Many recent works focus on certifying the robustness of learned neural networks under *any* attack. Approaches based on bounding the certified radius layer by layer using some convex relaxation methods have been proposed for certifying the robustness of neural networks (Wong & Kolter, 2018b; Gowal et al., 2018; Mirman et al., 2018; Dvijotham et al., 2018; Raghunathan et al., 2018; Zhang et al., 2020). However, such approaches are usually computationally expensive and have difficulties in scaling to deep models.

More recently, researchers found a new approach called randomized smoothing that can provide a certified robustness guarantee for general models. Lecuyer et al. (2018); Li et al. (2018); Cohen et al. (2019); Salman et al. (2019); Zhai et al. (2020) showed that if a Gaussian random noise is

added to the input layer, a certified guarantee on small $\ell_2$ perturbations can be computed for Gaussian smoothed classifiers. However, Yang et al. (2020); Blum et al. (2020); Kumar et al. (2020) showed that randomized smoothing cannot achieve nontrivial certified accuracy against more than $\Omega\left(\min\left(1, d^{1/p-1/2}\right)\right)$ radius for $\ell_p$ perturbations, where $d$ is the input dimension, therefore it cannot provide meaningful results for $\ell_\infty$ perturbations due to the curse of dimensionality.

Another line of more conservative certification approaches sought to bound the global Lipschitz constant of the neural network (Gouk et al., 2018; Tsuzuku et al., 2018; Anil et al., 2019; Cisse et al., 2017). However, the bounds are not tight (Cohen et al., 2019), and the final robustness performances are not as good as other certification methods.

# 3 $\ell_\infty$-DIST NETWORK AND ITS ROBUSTNESS GUARANTEE

## 3.1 PRELIMINARIES

Throughout this paper, we will use bold letters to denote vectors and otherwise scalars. Consider a standard classification task with an underlying data distribution $\mathcal{D}$ over pairs of examples $\boldsymbol{x} \in \mathcal{X}$ and corresponding labels $y \in \mathcal{Y} = \{1, 2, \cdots, M\}$. Usually $\mathcal{D}$ is unknown and we can only access a training set $\mathcal{T} = \{(\boldsymbol{x}_1, y_1), \cdots, (\boldsymbol{x}_n, y_n)\}$ in which $(\boldsymbol{x}_i, y_i)$ is *i.i.d.* drawn from $D$, $i = 1, 2, \cdots, n$. Let $f \in \mathcal{F}$ be the classifier of interest that maps any $\boldsymbol{x} \in \mathcal{X}$ to $\mathcal{Y}$. We call $\boldsymbol{x}' = \boldsymbol{x} + \boldsymbol{\delta}$ an *adversarial example* of $\boldsymbol{x}$ to classifier $f$ if $f$ can correctly classify $\boldsymbol{x}$ but assigns a different label to $\boldsymbol{x}'$. In real practice, the most commonly used setting is to consider the attack under $\epsilon$-bounded $\ell_\infty$ norm constraint, i.e., $\boldsymbol{\delta}$ satisfies $\|\boldsymbol{\delta}\|_\infty \leq \epsilon$, which is also called $\ell_\infty$ perturbations.

Our goal is to learn a model from $\mathcal{T}$ that can resist attacks at $(\boldsymbol{x}, y)$ with high probability over $(\boldsymbol{x}, y) \sim \mathcal{D}$ for any small $\ell_\infty$ perturbations. It relates to compute the radius of the largest $\ell_\infty$ ball centered at $\boldsymbol{x}$ in which $f$ does not change its prediction. This radius is called the *robust radius*, which is defined as (Zhai et al., 2020; Zhang et al., 2019a):

$$R(f; \boldsymbol{x}, y) = \begin{cases} \inf_{f(\boldsymbol{x}') \neq f(\boldsymbol{x})} \|\boldsymbol{x}' - \boldsymbol{x}\|_\infty \, , \text{ when } f(\boldsymbol{x}) = y \\ 0 \qquad\qquad\qquad , \text{ when } f(\boldsymbol{x}) \neq y \end{cases} \tag{1}$$

Unfortunately, computing the robust radius (1) of a classifier induced by a standard deep neural network is very difficult. For example, Weng et al. (2018) showed that computing the $\ell_p$ robust radius of a deep neural network is NP-hard for some specific $p$. Researchers then seek to derive a tight lower bound of $R(f; \boldsymbol{x}, y)$ for general $f$. This lower bound is called *certified radius* and we denote it as $CR(f; \boldsymbol{x}, y)$. The certified radius satisfies $0 \leq CR(f; \boldsymbol{x}, y) \leq R(f; \boldsymbol{x}, y)$ for any $f, \boldsymbol{x}, y$.

## 3.2 NETWORKS WITH $\ell_\infty$-DIST NEURONS

In this subsection, we propose the $\ell_\infty$-dist Neuron that is inherently Lipschitz with respect to $\ell_\infty$ norm. Using these neurons as building blocks, we then show how to obtain an $\ell_\infty$-Lipschitz neural networks dubbed $\ell_\infty$-dist Net.

Denote $\boldsymbol{z}$ as the input vector to a neuron. A standard neuron processes the input by first projecting $\boldsymbol{z}$ to a scalar value using a linear transformation, and then applying an non-linear activation function $\sigma$ on it, i.e., $\sigma(\boldsymbol{w}^\top \boldsymbol{z} + b)$. $\boldsymbol{w}$ and $b$ are parameters, and function $\sigma$ can be the sigmoid or ReLU activation. Unlike the previous design paradigm, we introduce a new type of neuron using $\ell_\infty$ distance as the basic operation, which we call $\ell_\infty$-dist neuron:

$$u(\boldsymbol{z}, \theta) = \|\boldsymbol{z} - \boldsymbol{w}\|_\infty + b, \tag{2}$$

where $\theta = \{\boldsymbol{w}, b\}$ is the parameter set. From the above equation, we can see that the $\ell_\infty$-dist neuron is non-linear as it calculates the $\ell_\infty$ distance between input $\boldsymbol{z}$ and parameter $\boldsymbol{w}$ with a bias term $b$. As a result, there is no need to further apply a non-linear activation function.

Without loss of generality, we study the properties of multi-layer perceptron (MLP) networks that are constructed using $\ell_\infty$-dist neurons. All theoretical results can be easily extended to other neural network architectures, such as convolutional neural nets. We use $\boldsymbol{x} \in \mathbb{R}^{d_{\text{input}}}$ to denote the input vector of an MLP network. A (MLP) network using $\ell_\infty$-dist neurons can be formally defined as follows.

**Definition 3.1.** *($\ell_\infty$-dist Net) Denote $\boldsymbol{g}$ as an MLP network which takes $\boldsymbol{x}^{(1)} \triangleq \boldsymbol{x} \in \mathbb{R}^{d_{\mathrm{input}}}$ as input. Assume there are $L$ hidden layers and the $l$-th hidden layer contains $d_l$ hidden units. We call $\boldsymbol{g}$ is an MLP network constructed by $\ell_\infty$-dist neurons, if the $k$-th unit in the $l$-th hidden layer $x_k^{(l+1)}$ is computed by*

$$x_k^{(l+1)} = u(\boldsymbol{x}^{(l)}, \theta^{(l,k)}) = \|\boldsymbol{x}^{(l)} - \boldsymbol{w}^{(l,k)}\|_\infty + b^{(l,k)}, 0 < l \leq L, 0 < k \leq d_l,$$

*where $\boldsymbol{x}^{(l)} = (x_1^{(l)}, x_2^{(l)}, \cdots, x_{d_{l-1}}^{(l)})$ is the input vector to the $l$-th hidden layer.*

For simplicity, we will call an MLP network constructed by $\ell_\infty$-dist neurons as $\ell_\infty$-*dist Net*. For classification tasks, the dimensionality of the output of $\boldsymbol{g}$ matches the number of categories, i.e., $M$. We write $\boldsymbol{g}(\boldsymbol{x}) = (g_1(\boldsymbol{x}), g_2(\boldsymbol{x}), \cdots, g_M(\boldsymbol{x}))$ and define the predictor $f(\boldsymbol{x}) = \arg\max_{i \in [M]} g_i(\boldsymbol{x})$. Note that $\boldsymbol{g}(\boldsymbol{x})$ can be used as the output logits the same as in any other networks, so we can apply any standard loss function on the $\ell_\infty$-dist Net, such as the cross-entropy loss or hinge loss.

We will further show that the $\ell_\infty$-dist neuron and the neural networks constructed using it have nice theoretical properties to control the robustness of the model. For completeness, we first introduce the definition of Lipschitz functions as below.

**Definition 3.2.** *(Lipschitz Function) A function $\boldsymbol{g}(\boldsymbol{z}) : \mathbb{R}^m \to \mathbb{R}^n$ is called $\lambda$-Lipschitz with respect to $\ell_p$ norm $\|\cdot\|_p$, if for any $\boldsymbol{z}_1, \boldsymbol{z}_2$, the following holds:*

$$\|\boldsymbol{g}(\boldsymbol{z}_1) - \boldsymbol{g}(\boldsymbol{z}_2)\|_p \leq \lambda\|\boldsymbol{z}_1 - \boldsymbol{z}_2\|_p$$

It is straightforward to see that the $\ell_\infty$-dist neuron is 1-Lipschitz with respect to $\ell_\infty$ norm.

### 3.3 LIPSCHITZ AND ROBUSTNESS FACTS ABOUT $\ell_\infty$-DIST NETS

In this subsection, we introduce two straightforward but important facts about $\ell_\infty$-dist Nets. We first show that $\ell_\infty$-dist Nets are still 1-Lipschitz with respect to $\ell_\infty$ norm, and then derive the certified robustness of the model based on this property.

**Fact 3.1.** *Any $\ell_\infty$-dist Net $\boldsymbol{g}(\cdot)$ is 1-Lipschitz with respect to $\ell_\infty$ norm, i.e., for any $\boldsymbol{x}_1, \boldsymbol{x}_2 \in \mathbb{R}^{d_{\mathrm{input}}}$, we have $\|\boldsymbol{g}(\boldsymbol{x}_1) - \boldsymbol{g}(\boldsymbol{x}_2)\|_\infty \leq \|\boldsymbol{x}_1 - \boldsymbol{x}_2\|_\infty$.*

*Proof.* It's easy to check that every operation $u(\boldsymbol{x}^{(l)}, \theta^{(l,k)})$ is 1-Lipschitz, and therefore the mapping from one layer to the next $\boldsymbol{x}^{(l)} \to \boldsymbol{x}^{(l+1)}$ is 1-Lipschitz. Then we have for any $\boldsymbol{x}_1, \boldsymbol{x}_2 \in \mathbb{R}^{d_{input}}$, $\|\boldsymbol{g}(\boldsymbol{x}_1) - \boldsymbol{g}(\boldsymbol{x}_2)\|_\infty \leq \|\boldsymbol{x}_1 - \boldsymbol{x}_2\|_\infty$. $\square$

Since $\boldsymbol{g}$ is 1-Lipschitz with respect to $\ell_\infty$ norm, if the perturbation over $\boldsymbol{x}$ is rather small, the change of the output can be bounded and the label of the perturbed data $\boldsymbol{x}'$ will not change as long as $\arg\max_{i \in [M]} g_i(\boldsymbol{x}) = \arg\max_{i \in [M]} g_i(\boldsymbol{x}')$, which directly bounds the certified radius.

**Fact 3.2.** *Given model $f(\boldsymbol{x}) = \arg\max_{i \in [M]} g_i(\boldsymbol{x})$ defined as above, and $\boldsymbol{x}$ is correctly classified as $y$. We define $\mathrm{margin}(\boldsymbol{x}; \boldsymbol{g})$ as the difference between the largest and second-largest elements of $\boldsymbol{g}(\boldsymbol{x})$, then for any $\boldsymbol{x}'$ satisfying $\|\boldsymbol{x} - \boldsymbol{x}'\|_\infty \leq \mathrm{margin}(\boldsymbol{x}; \boldsymbol{g})/2$, we have that $f(\boldsymbol{x}) = f(\boldsymbol{x}')$ and:*

$$CR(f, \boldsymbol{x}, y) \geq \frac{\mathrm{margin}(\boldsymbol{x}; \boldsymbol{g})}{2} \tag{3}$$

*Proof.* Since $\boldsymbol{g}(\boldsymbol{x})$ is 1-Lipschitz, each element of $\boldsymbol{g}(\boldsymbol{x})$ can move at most $\frac{\mathrm{margin}(\boldsymbol{x}; \boldsymbol{g})}{2}$ when $x$ changes to $x'$, therefore the largest element will remain the same. $\square$

Using this bound, we can certify the robustness of an $\ell_\infty$-dist Net of *any size* under $\ell_\infty$ norm with little computational cost (only a forward pass). In contrast, existing certified methods suffer from either poor scalability (methods based on convex relaxation) or curse of dimensionality (randomized smoothing), and lack efficiency as well.

## 4 Expressive power and Robust Generalization of $\ell_\infty$-dist Net

The expressive power of a model family and its generalization are two central topics in machine learning. In the previous section, we show that an $\ell_\infty$-dist Net is 1-Lipschitz with respect to $\ell_\infty$ norm. Then it's natural to ask whether the designed neural network can approximate *any* 1-Lipschitz function (with respect to $\ell_\infty$ norm) and whether we can give theoretical guarantees on the robust test error of a learned model.

In this section, we give affirmative answers to both questions. We will first prove a Lipschitz-universal approximation theorem for $\ell_\infty$-dist Net using a structured approach, then give upper bounds of robust test error based on the Rademacher complexity of the model family. Without loss of any generality, we consider *binary classification* problems and assume the output dimension is 1. All the omitted proofs in this section can be found in the appendix.

### 4.1 Lipschitz-Universal Approximation of $\ell_\infty$-dist Net

In this subsection, we show that $\ell_\infty$-dist Nets with bounded width can approximate *any* 1-Lipschitz function (with respect to $\ell_\infty$ norm), formalized in the following theorem:

**Theorem 1.** *For any 1-Lipschitz function $\tilde{g}(\boldsymbol{x})$ (with respect to $\ell_\infty$ norm) on a bounded domain $\mathbb{K} \in R^{d_{\mathrm{input}}}$ and any $\epsilon > 0$, there exists an $\ell_\infty$-dist Net $g(\boldsymbol{x})$ with width no more than $d_{input} + 2$, such that for all $\boldsymbol{x} \in \mathbb{K}$, we have*

$$\|g(\boldsymbol{x}) - \tilde{g}(\boldsymbol{x})\|_\infty \leq \epsilon. \tag{4}$$

To prove Theorem 1, we need the following key lemma:

**Lemma 4.1.** *For any 1-Lipschitz function $f(\boldsymbol{x})$ (with respect to $\ell_\infty$ norm) on a bounded domain $\mathbb{K} \in R^n$, and any $\epsilon > 0$, there exists a finite number of functions $f_i(\boldsymbol{x})$ such that for all $\boldsymbol{x} \in \mathbb{K}$*

$$\max_i f_i(\boldsymbol{x}) \leq f(\boldsymbol{x}) \leq \max_i f_i(\boldsymbol{x}) + \epsilon,$$

*where each $f_i(\boldsymbol{x})$ has the following form*

$$f_i(\boldsymbol{x}) = \min\{x_1 - w_1^{(i)}, w_1^{(i)} - x_1, x_2 - w_2^{(i)}, ..., w_n^{(i)} - x_n\} + b_i.$$

Lemma 4.1 "decomposes" any target 1-Lipschitz function into simple "base functions", which will serve as building blocks in proving the main theorem. Lu et al. (2017) first proved such universal approximation theorem for width-bounded ReLU networks by constructing a special network that approximates the target function by "sum of cubes". For $\ell_\infty$-dist Net, such an approach cannot be directly applied as the summation will break the Lipschitz property. We employ a novel "max of pyramids" construction to overcome the issue. The key idea is to approximate the target function from below using the maximum of many "pyramid-like" basic 1-Lipschitz functions.

Theorem 1 implies that an $\ell_\infty$-dist Net can approximate any 1-Lipschitz function with respect to $\ell_\infty$ norm on a compact set, using only finite width which is barely larger than the input dimension. Combining with the fact that an $\ell_\infty$-dist Net is 1-Lipschitz, we conclude that $\ell_\infty$-dist Nets is a good class of model to approximate 1-Lipschitz functions.

### 4.2 Bounding robust test error of $\ell_\infty$-dist Net

Good robust generalization ability, i.e. small empirical (margin) error implies small robust test error, is as important as strong expressive power. In this subsection, we give upper bound for the *robust test error* of $\ell_\infty$-dist Net.

Consider the following classification problem: let $(\boldsymbol{x}, y)$ be an instance-label couple where $\boldsymbol{x} \in \mathbb{K}$ and $y \in \{1, -1\}$ and denote $\mathcal{D}$ as the distribution of $(\boldsymbol{x}, y)$. For a function $g(\boldsymbol{x}) : \mathbb{R}^{d_{\mathrm{input}}} \to \mathbb{R}$, we use $\mathrm{sign}(g(\boldsymbol{x}))$ as the classifier. The $r$-robust test error $\gamma_r$ of a classifier $g$ is defined as

$$\gamma_r = \mathbb{E}_{\mathbb{D}}\left[\sup_{\|\boldsymbol{x}' - \boldsymbol{x}\|_\infty \leq r} \mathbb{I}_{yg(\boldsymbol{x}') \leq 0}\right]$$

We can upper bound $\gamma_r$ by the margin error on training data and the size of network:

**Theorem 2.** *Let $\mathbb{F}$ denote the set of all $g$ represented by an $\ell_\infty$-dist Net with width $w$ and depth $L$, for every $t > 0$, with probability at least $1 - 2e^{-2t^2}$ over the random drawing of $n$ samples, for all $r > 0$ and $g \in \mathbb{F}$ we have that*

$$\gamma_r \leq \inf_{\delta \in (0,1]} \left[ \underbrace{\frac{1}{n} \sum_{i=1}^{n} \mathbb{I}_{y_i g(\boldsymbol{x}_i) \leq \delta + r}}_{\text{large training margin}} + \underbrace{\tilde{O}\left(\frac{Lw^2}{\delta\sqrt{n}}\right)}_{\text{network size}} + \left(\frac{\log\log_2(\frac{2}{\delta})}{n}\right)^{\frac{1}{2}} \right] + \frac{t}{\sqrt{n}}. \tag{5}$$

Theorem 2 provides a theoretical guarantee on the adversarial robust test error: when a large margin classifier is found on training data, and the size of the $\ell_\infty$-dist Net is not too large, then with high probability the model can generalize well in terms of adversarial robustness.

We prove the theorem in two steps. One step is to provide a margin bound to control the gap between standard training error and standard test error. In this step, we use Rademacher complexity $R(\mathbb{F})$ of a hypothesis set $\mathbb{F}$, where the test error $\beta_r$ with $r$ as margin is defined as: $\beta_r = \mathbb{E}_\mathcal{D}\left[\mathbb{I}_{yg(\boldsymbol{x}) \leq r}\right]$. The other step is to bound the gap between test error and robust test error. The two steps are provided in the following two lemmas.

**Lemma 4.2.** *Let $\mathbb{F}$ be a hypothesis class, then for any $t > 0$,*

$$\mathbb{P}\left(\exists g \in \mathbb{F}: \beta_r > \inf_{\delta \in (0,1]} \left[\frac{1}{n}\sum_{i=1}^{n}\mathbb{I}_{y_i g(\boldsymbol{x}_i) \leq \delta + r} + \frac{48}{\delta}R_n(\mathbb{F}) + \left(\frac{\log\log_2(\frac{2}{\delta})}{n}\right)^{\frac{1}{2}}\right] + \frac{t}{\sqrt{n}}\right) \leq 2e^{-2t^2}.$$

**Lemma 4.3.** *The $r$-robust test error $\gamma_r$ is no larger than the test error $\beta_r$, i.e., $\gamma_r \leq \beta_r$.*

## 5 TRAINING $\ell_\infty$-DIST NET

Motivated by the theoretical analysis in the previous sections, we would like to find a large margin solution on the training data. Therefore we use the standard hinge loss as the objective function:

$$L(g, \mathcal{T}) = \frac{1}{n}\sum_{i=1}^{n}\max\left\{0, t + \max_{y_i' \neq y_i} \boldsymbol{g}(\boldsymbol{x}_i)_{y_i'} - \boldsymbol{g}(\boldsymbol{x}_i)_{y_i}\right\} \tag{6}$$

where $t$ controls the desired margin and should be slightly larger than twice of the desired robustness level according to Eqn. 3. It can be easily seen that $L(g, \mathcal{T})$ is a continuous function of the network parameters and is differentiable almost everywhere. Therefore any gradient based optimization method can be used to train an $\ell_\infty$-dist Net.

However, the optimization is not easy. We find that directly using SGD or Adam optimizer *fails* to train the network well. In fact, the gradient of the $\ell_\infty$ norm (i.e., $\|\boldsymbol{z} - \boldsymbol{w}\|_\infty$) with respect to $\boldsymbol{w}$ is very sparse, which makes the optimization inefficient. In this section, we list a few important optimization tricks which we find is effective in practice.

**Normalization and parameter initialization** Batch Normalization (Ioffe & Szegedy, 2015), which shifts and scales feature values in each layer, is shown to be one of the most important components in training deep neural networks. However, in $\ell_\infty$-dist Net, if we directly apply batch normalization, the Lipschitz constant of the network will change due to the scaling operation, and the robustness of the trained model cannot be guaranteed. To leverage the advantages of BatchNorm, we only apply the shift operation that shifts the output of each layer using the mean over the current batch during training. Similar to BatchNorm, we use the running mean during inference. Note that in inference, the running mean serves as an additional bias term in $\ell_\infty$-dist neuron, which does not affect the Lipschitz constant. Finally, we do not use affine transformation which is typically used in BatchNorm.

We apply the shift operation in all intermediate layers after calculating the $\ell_\infty$-dist norm, but not the final layer. As a result, we remove the bias term in the $\ell_\infty$-dist neuron in all intermediate layers. The bias in the last layer is initialized to zero. As for the weight initialization, weights and inputs should have the same scale due to the $\ell_\infty$ distance operation. Therefore we initialize all weights from standard Gaussian distribution since the inputs are normalized to have unit variance.

**Smoothed approximated gradients.** We find that training an $\ell_\infty$-dist from scratch is usually inefficient, and the model can easily be stuck in some bad local minima. To obtain a well-optimized model, we relax the $\ell_\infty$ distance by $\ell_p$ distance for each neuron, to get an approximate and non-sparse gradient of the parameters (Chen et al., 2020). During training, we set $p$ to be a small value in the beginning, and linearly increase it in each iteration to reach a large value. When the training finishes, we freeze the model parameter and set $p$ to $\infty$ as our final model.

# 6 EXPERIMENT

## 6.1 EXPERIMENTAL SETTING

**Model details and training configurations.** We train the $\ell_\infty$-dist Net on two benchmark datasets: MNIST and Fashion-MNIST. For both tasks, we use a 4-layer $\ell_\infty$-dist Net. Each hidden layer has 4096 neurons. Normalization is applied between each intermediate layer. The batch size is set to 512. Random-crop data augmentation and image standardization are used during training. We train the network using Adam optimizer with hyper-parameters $\beta_1 = 0.9$ and $\beta_2 = 0.99$. The training procedure is as follows: First, we relax the $\ell_\infty$-dist Net to $\ell_p$-dist Net with $p = 6$ and train the model parameters for 30 epochs. Then we gradually increase $p$ from 6 to 100 in the next 170 epochs. Finally, we fix $p = 100$ and train the last 50 epochs. The learning rate in the first and second parts is set to be 0.015 and is divided by 5.0 in the final 50 epochs. We use hinge loss with parameter $t = 0.9$. For the Fashion-MNIST dataset, we use exactly *the same model and the same hyper-parameters* as MNIST, except for the hinge loss parameter $t = 0.4$. Adjusting models and hyper-parameters may further improve the results.

**Evaluation.** Following Wong & Kolter (2018b); Gowal et al. (2018); Madry et al. (2017); Zhang et al. (2019b), we test the robustness of $\ell_\infty$-dist Net under $\ell_\infty$ radius 0.3 on MNIST and 0.1 on Fashion-MNIST. We use two evaluation metrics to measure the robustness of the model. We first evaluate the robust test accuracy under the Projected Gradient Descent (PGD) attack (Madry et al., 2017). Following standard practice, we set the number of steps of the PGD attack to be 20 and set the step size to be 0.01. We also calculate the certified radius for each sample using Eq. 3, and check the percentage of test samples that can be certified to be robust within the chosen radius.

Table 1: Comparison of different methods on MNIST under $\ell_\infty$ radius 0.3.

| Method | w/o Adv Train | Scalable | Standard Acc | Robust Acc | Certified Acc |
|---|---|---|---|---|---|
| Natural | ✓ | ✓ | 99.3 | $\approx 0$ | $\approx 0$ |
| IBP | ✗ | ✗ | 98.34 | 93.88 | 91.95 |
| CROWN-IBP | ✗ | ✗ | 98.18 | 93.95 | 92.98 |
| GroupSort | ✓ | ✗ | $\approx 97.0$ | $\approx 32.0$ | $\approx 2.0$ |
| $\ell_\infty$-dist Net | ✓ | ✓ | 98.61 | 93.78 | 91.59 |

Table 2: Comparison of different methods on Fashion-MNIST under $\ell_\infty$ radius 0.1.

| Method | w/o Adv Train | Scalable | Standard Acc | Robust Acc | Certified Acc |
|---|---|---|---|---|---|
| Natural | ✓ | ✓ | 94.0 | $\approx 0$ | $\approx 0$ |
| CAP | ✗ | ✗ | 78.27 | 68.37 | 65.47 |
| IBP | ✗ | ✗ | - | - | 76.51 |
| $\ell_\infty$-dist Net | ✓ | ✓ | 87.46 | 75.44 | 73.23 |

**Baselines.** In Table 1 and 2, we compare $\ell_\infty$-dist Net with several baselines, i.e., standard training on standard neural network (abbreviated as Natural in the table), CAP (Wong & Kolter, 2018b), IBP (Gowal et al., 2018), CROWN-IBP (Zhang et al., 2019b), GroupSort Network (Anil et al., 2019). Randomized Smoothing (Cohen et al., 2019; Salman et al., 2019; Zhai et al., 2020) is another strong baseline when considering the robustness with respect to $\ell_2$ perturbation. However, as discussed in the related work section, it cannot achieve nontrivial certified accuracy against more than $\Omega\left(\min\left(1, d^{-1/2}\right)\right)$ radius for $\ell_\infty$ perturbation where $d$ is the input dimension, which implies randomized smoothing cannot obtain good certification under $\ell_\infty$ norm. We report the performances

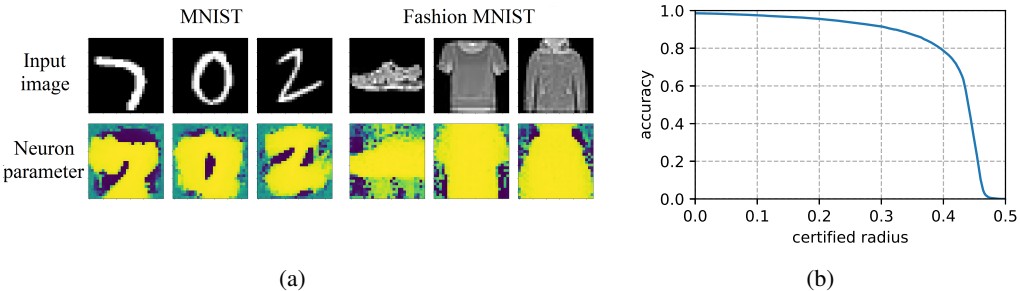

(a)                                                            (b)

Figure 1: (a) We sample some neurons from the first layer and list the parameters $w$ together with some input data on MNIST and Fashion-MNIST. It can be seen that on both tasks, the parameters (first row) look very similar to the data (second row) in some categories. This shows that the neuron captures the shape information, which is a meaningful feature and useful for classification. (b) The radius-(certified accuracy) curve of our trained model on MNIST dataset.

(picked from the original papers) together with other properties of these methods, e.g., whether these methods use adversarial training (abbreviated as Adv Train in the table); are these methods scalable to large neural networks.

## 6.2 EXPERIMENTAL RESULTS

We list the performances of $\ell_\infty$-dist Net in Table 1 and 2. In these tables, we use "Standard Acc", "Robust Acc" and "Certified Acc" as abbreviations of standard (clean) test accuracy, robust test accuracy under PGD attack and certified robust test accuracy. All the numbers are reported in percentage.

From Table 1 and 2, we find that $\ell_\infty$-dist Net can achieve comparative or better robustness than other baselines while maintaining high standard accuracy. Specifically, on MNIST, among the methods that do not need adversarial training or can be scaled to large networks, $\ell_\infty$-dist Net has the best robustness and achieves more than 93% robust test accuracy and more than 91% certified test accuracy. These performances are also comparative with those methods that train the model using adversarial training and calculate the certified radius with huge computational cost (Wong & Kolter, 2018b; Gowal et al., 2018; Zhang et al., 2019b). We also plot the radius-(certified accuracy) curve on MNIST in Figure 1(b). Results on Fashion-MNIST are similar.

To understand how the $\ell_\infty$-dist Net extracts information from the data, we sample some neurons in the first layer and record its parameter $w$. We then visualize $w$ together with some input data on both tasks. Interestingly, in Figure 1(a), we can see that the parameters $w$ (first row) "look similar" to the input data (second row). This phenomenon is because we use $\ell_\infty$ distance in the network to obtain effective signals for classification. Then the neurons seek to find typical patterns that can differentiate one category to others in terms of $\ell_\infty$ distance, e.g., the shape of the objects, which is quite different from the feature extractor in standard neural networks.

## 7 CONCLUSION

In this paper, we design a novel neuron that uses $\ell_\infty$ distance as its basic operation. We show that the $\ell_\infty$-dist neuron is naturally a 1-Lipschitz function with respect to the $\ell_\infty$ norm and the neural networks constructed with $\ell_\infty$-dist neuron ($\ell_\infty$-dist Nets) enjoy the same property. This directly provides a theoretical guarantee of the certified robustness based on the margin of the prediction outputs. We further formally analyze the expressiveness power and the robust generalization ability of the network. Preliminary experiments show promising results on MNIST and Fashion-MNIST datasets. As future work, we will conduct experiments of the $\ell_\infty$-dist Net on more datasets (CIFAR10 and ImageNet) and extend it to modern networks, such as deep convolutional networks with residual connections.

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

# A   Appendix

## A.1   Proof of Lemma 4.1

*Proof.* Without loss of generality we may assume $\mathbb{K} \in [0,1]^n$. Consider the set $\mathbb{S}$ consisting of all points $\frac{\epsilon}{2}(N_1,...,N_n)$ where $N_j$ are integers, we can write $\mathbb{S} \cap \mathbb{K} = \{\boldsymbol{w}^{(1)},...,\boldsymbol{w}^{(m)}\}$ since it's a finite set. $\forall \boldsymbol{w}^{(i)} \in \mathbb{S} \cap \mathbb{K}$, we define the corresponding $f_i(\boldsymbol{x})$ as follows

$$f_i(\boldsymbol{x}) = \min\{x_1 - w_1^{(i)}, w_1^{(i)} - x_1, x_2 - w_2^{(i)}, ..., w_n^{(i)} - x_n\} + f(\boldsymbol{w}^{(i)}) \tag{7}$$

Obviously $\forall \boldsymbol{x} \in \mathbb{K}$ we have that $f(\boldsymbol{x}) \geq f_i(\boldsymbol{x})$, thus $\max_i f_i(\boldsymbol{x}) \leq f(\boldsymbol{x})$ holds as a direct consequence.

On the other hand, $\forall \boldsymbol{x} \in \mathbb{K}$ there exists its 'neighbour' $\boldsymbol{w}^{(j)} \in \mathbb{S} \cap \mathbb{K}$ such that $\|\boldsymbol{x} - \boldsymbol{w}^{(j)}\|_\infty \leq \frac{\epsilon}{2}$, therefore by using the Lipschitz properties of both $f(\boldsymbol{x})$ and $f_j(\boldsymbol{x})$, we have that

$$f(\boldsymbol{x}) \leq f(\boldsymbol{w}^{(j)}) + \frac{\epsilon}{2} = f_j(\boldsymbol{w}^{(j)}) + \frac{\epsilon}{2} \leq f_j(\boldsymbol{x}) + \epsilon \tag{8}$$

Since $\max_i f_i(\boldsymbol{x}) \geq f_j(\boldsymbol{x})$, we conclude our proof.   □

## A.2   Proof of Theorem 1

*Proof.* By lemma 4.1, there exists a finite number of functions $f_i(\boldsymbol{x})$ $(i = 1,...,m)$ such that $\forall \boldsymbol{x} \in \mathbb{K}$

$$\max_i f_i(\boldsymbol{x}) \leq \tilde{g}(\boldsymbol{x}) \leq \max_i f_i(\boldsymbol{x}) + \epsilon \tag{9}$$

where each $f_i(\boldsymbol{x})$ has the form

$$f_i(\boldsymbol{x}) = \min\{x_1 - w_1^{(i)}, w_1^{(i)} - x_1, x_2 - w_2^{(i)}, ..., w_n^{(i)} - x_n\} + \tilde{g}(\boldsymbol{w}^{(i)}) \tag{10}$$

The high-level idea of the proof is very simple: among width $d_{\text{input}} + 2$, we allocate $d_{\text{input}}$ neurons each layer to keep the information of $\boldsymbol{x}$, one to calculate each $\tilde{f}_i(\boldsymbol{x})$ one after another and the last neuron calculating the maximum of $f_i(\boldsymbol{x})$ accumulated.

To simplify the proof, we would first introduce three general basic maps which can be realized at a single unit, then illustrate how to represent $\max_i f_i(\boldsymbol{x})$ by combing these basic maps.

Let's assume for now that any input to any unit in the whole network has its $\ell_\infty$ norm upper bounded by a large constant $C$, we will come back later to determine this value and prove its validity.

**Proposition A.1.** $\forall j, k, p$ and constant $w, c$, the following base functions are realizable at the $k$th unit in the $l$th hidden layer:

1, the projection map:

$$u(\boldsymbol{x}^{(l)}, \theta^{(l,k)}) = x_j^{(l)} + c \tag{11}$$

2, the negation map:

$$u(\boldsymbol{x}^{(l)}, \theta^{(l,k)}) = -x_j^{(l)} + c \tag{12}$$

3, the maximum map:

$$u(\boldsymbol{x}^{(l)}, \theta^{(l,k)}) = \max\{x_j^{(l)} + w, x_p^{(l)}\} + c, u(\boldsymbol{x}^{(l)}, \theta^{(l,k)}) = \max\{-x_j^{(l)} + w, x_p^{(l)}\} + c \tag{13}$$

*Proof.* 1, the projection map: Setting $u(\boldsymbol{x}^{(l)}, \theta^{(l,k)})$ as follows

$$u(\boldsymbol{x}^{(l)}, \theta^{(l,k)}) = \|(x_1^{(l)}, ..., x_j^{(l)} + 2C, ..., x_n^{(l)})\|_\infty - 2C + c \tag{14}$$

2, the negation map: Setting $u(\boldsymbol{x}^{(l)}, \theta^{(l,k)})$ as follows

$$u(\boldsymbol{x}^{(l)}, \theta^{(l,k)}) = \|(x_1^{(l)}, ..., x_j^{(l)} - 2C, ..., x_n^{(l)})\|_\infty - 2C + c \tag{15}$$

3, the maximum map: Setting $u(\boldsymbol{x}^{(l)}, \theta^{(l,k)})$ as follows

$$u(\boldsymbol{x}^{(l)}, \theta^{(l,k)}) = \|(x_1^{(l)}, ..., x_j^{(l)} + w + 2C, ..., x_p^{(l)} + 2C, ..., x_n^{(l)})\|_\infty - 2C + c \tag{16}$$

$$u(\boldsymbol{x}^{(l)}, \theta^{(l,k)}) = \|(x_1^{(l)}, ..., -x_j^{(l)} + w + 2C, ..., x_p^{(l)} + 2C, ..., x_n^{(l)})\|_\infty - 2C + c \tag{17}$$

$\square$

With three basic maps in hand, we are prepared to construct our network. Using proposition A.1, the first hidden layer realizes $u(\boldsymbol{x}, \theta^{(1,k)}) = x_k$ for $k = 1, ..., d_{\text{input}}$. The rest two units can be arbitrary, we set both to be $x_1$.

By proposition A.1, throughout the whole network, we can set $u(\boldsymbol{x}^{(l)}, \theta^{(l,k)}) = x_k$ for all $l$ and $k = 1, ..., n$. Notice that $f_i(\boldsymbol{x})$ can be rewritten as

$$f_i(\boldsymbol{x}) = -\max\{x_1 - w_1^{(i)}, \max\{w_1^{(i)} - x_1, \max\{..., w_n^{(i)} - x_n\}...\}\} + \tilde{g}(\boldsymbol{w}^{(i)}) \tag{18}$$

Using the maximum map recurrently while keeping other units unchanged with the projection map, we can utilize the unit $u(\boldsymbol{x}^{(l)}, \theta^{(l,d_{\text{input}}+1)})$ to realize one $f_i(\boldsymbol{x})$ at a time. Again by the use of maximum map, the last unit $u(\boldsymbol{x}^{(l)}, \theta^{(l,d_{\text{input}}+2)})$ will recurrently calculate (initializing with $\max\{f_1(\boldsymbol{x})\} = f_1(\boldsymbol{x})$)

$$\max_i f_i(\boldsymbol{x}) = \max\{f_m(\boldsymbol{x}), \max\{..., \max\{f_1(\boldsymbol{x})\}...\}\} \tag{19}$$

using only finite depth, say $L$, then the network outputs $g(\boldsymbol{x}) = u(\boldsymbol{x}^{(L)}, \theta^{(L,1)}) = u(\boldsymbol{x}^{(L-1)}, \theta^{(L-1,d_{\text{input}}+2)}) = \max_i f_i(\boldsymbol{x})$ as desired. We are only left with deciding a valid value for $C$. Because $\mathbb{K}$ is bounded and $\tilde{g}(\boldsymbol{x})$ is continuous, there exists constants $C_1, C_2$ such that $\forall \boldsymbol{x} \in \mathbb{K}$, $\|\boldsymbol{x}\|_\infty \leq C_1$ and $|\tilde{g}(\boldsymbol{x})| \leq C_2$, it's easy to verify that $C = 2C_1 + C_2$ is valid.

$\square$

### A.3 PROOF OF LEMMA 4.2

First we give a quick revisit on Rademacher complexity and its properties.

**Rademacher Complexity** Given a sample $X_n = \{\boldsymbol{x}_1, ..., \boldsymbol{x}_n\} \in \mathbb{K}^n$, and a real-valued function class $\mathbb{F}$ on $\mathbb{K}$, the Rademacher complexity of $\mathbb{F}$ is defined as

$$R_n(\mathbb{F}) = \mathbb{E}_{X_n} \left( \frac{1}{n} \mathbb{E}_\sigma \left[ \sup_{f \in \mathbb{F}} \sum_{i=1}^n \sigma_i f(\boldsymbol{x}_i) \right] \right)$$

where $\sigma_i$ are drawn from the Rademacher distribution independently, i.e. $P(\sigma_i = 1) = P(\sigma_i = -1) = \frac{1}{2}$. It's worth noting that $\forall r, R_n(\mathbb{F}) = R_n(\mathbb{F} \oplus r)$ where $\mathbb{F} \oplus r = \{f + r | f \in \mathbb{F}\}$.

It's well known (using Massart's Lemma) that Rademacher complexity can be bounded by VC dimension:

$$R_n(\mathbb{F}) \leq \sqrt{\frac{2VCdim(\mathbb{F}) \log \frac{en}{VCdim(\mathbb{F})}}{n}} \tag{20}$$

*Proof.* Denote the hypothesis set with $\mathbb{F}$ and its Rademacher complexity on training data with $R_n(\mathbb{F})$, we can upper bound the generalization error when a large margin solution is found on training data:

**Lemma A.1.** *(Theorem 11 in Koltchinskii et al. (2002)) For all $t > 0$,*

$$P \left( \exists g \in \mathbb{F} : \beta_0 > \inf_{\delta \in (0,1]} \left[ \frac{1}{n} \sum_{i=1}^n \mathbb{I}_{y_i g(\boldsymbol{x}_i) \leq \delta} + \frac{48}{\delta} R_n(\mathbb{F}) + \left( \frac{\log \log_2(\frac{2}{\delta})}{n} \right)^{\frac{1}{2}} \right] + \frac{t}{\sqrt{n}} \right) \leq 2e^{-2t^2}$$

To further generalize lemma A.1 to $\beta_r$ with $r > 0$, we use the fact that Rademacher complexity remain unchanged if the same constant $r$ is added to all functions in $\mathbb{F}$. Lemma 4.2 is a direct consequence by replacing $m_f$ by $m_f - r$ at the end of the proof of theorem 11 in Koltchinskii et al. (2002), where it plugs $m_f$ into theorem 2 in Koltchinskii et al. (2002):

$\square$

## A.4 Proof of Lemma 4.3

*Proof.* $yg(\boldsymbol{x}) > r$ implies $\inf_{\|x'-x\|_\infty \leq r} yg(\boldsymbol{x}') > 0$ because $g(\boldsymbol{x})$ is 1-Lipschitz, therefore

$$\mathbb{E}_\mathcal{D} \left[ \mathbb{I}_{yg(\boldsymbol{x}) > r} \right] \leq \mathbb{E}_\mathcal{D} \left[ \inf_{\|x'-x\|_\infty \leq r} \mathbb{I}_{yg(\boldsymbol{x}') > 0} \right] \leq \mathbb{E}_\mathcal{D} \left[ \sup_{\|x'-x\|_\infty \leq r} \mathbb{I}_{yg(\boldsymbol{x}') > 0} \right]$$

and so $1 - \beta_r \leq 1 - \gamma_r$ which concludes the proof. $\square$

## A.5 Proof of Theorem 2

*Proof.* There are generalization bounds like (Luxburg & Bousquet, 2004), which involves only the Lipschitz property. Unfortunately, in those bounds, the dependence on sample size is of order $n^{\frac{-1}{d_{\text{input}}}}$, which suffers from the curse of dimensionality. To derive a more meaningful bound, we take the network size into consideration as well.

We can bound the VC dimension of $\ell_\infty$-dist Net by reducing a given $\ell_\infty$-dist Net network to a fully-connected ReLU network. We first introduce the VC bound for fully-connected neural networks with ReLU activation borrowed from Bartlett et al. (2019):

**Lemma A.2.** *(Theorem 6 in Bartlett et al. (2019)) Consider a fully-connected ReLU network architecture $F$ with input dimension $d$, width $w \geq d$ and depth (number of hidden layers) $L$, then its VC dimension satisfies:*

$$VCdim(F) = \tilde{O}(L^2 w^2) \tag{21}$$

The following lemma shows how to calculate $\ell_\infty$ distance using a fully-connected ReLU network architecture.

**Lemma A.3.** $\forall \boldsymbol{w} \in R^d$, *there exists a fully-connected ReLU network $h$ with width $O(d)$ and depth $O(\log d)$ such that $h(\boldsymbol{x}) = \|\boldsymbol{x} - \boldsymbol{w}\|_\infty$*

*Proof.* The proof is by construction. Rewrite $\|\boldsymbol{x} - \boldsymbol{w}\|_\infty$ as

$$\max\{x_1 - w_1, w_1 - x_1, ..., x_d - w_d, w_d - x_d\}$$

which is a maximum of $2d$ items. Notice that $\max\{x, y\} = ReLU(x-y) + ReLU(y) - ReLU(-y)$, we can use $3d$ neurons in the first hidden layer so that the input to the second hidden layer are $\max\{x_i - w_i, w_i - x_i\}$, in all $d$ items. Repeating this procedure which cuts the number of items within maximum by half, within $O(\log d)$ hidden layers this network finally outputs $\|\boldsymbol{x} - \boldsymbol{w}\|_\infty$ as desired. $\qquad\square$

The VC bound of $\ell_\infty$-dist Net is formalized by the following lemma:

**Lemma A.4.** *Consider an $\ell_\infty$-dist Net architecture $F$ with input dimension $d$, width $w \geq d$ and depth (number of hidden layers) $L$, then its VC dimension satisfies:*

$$VCdim(F) = \tilde{O}(L^2 w^4) \tag{22}$$

*Proof.* By lemma A.3, each neuron in the $\ell_\infty$-dist Net can be replaced by a fully-connected ReLU subnetwork with width $O(w)$ and depth $O(\log w)$. Therefore a fully-connected ReLU network architecture $G$ with width $O(w^2)$ and depth $O(L \log w)$ can realize any function represented by the $\ell_\infty$-dist Net when parameters vary. Remember that VC dimension is monotone under the ordering of set inclusion, we conclude that such $\ell_\infty$-dist Net architecture $F$ has VC dimension no more than that of $G$ which equals $\tilde{O}(L^2 w^4)$ by lemma A.2. $\qquad\square$

Finally, Theorem 2 is a direct consequence by combing Lemmas 4.2, 4.3, A.3, A.2 and A.4. $\qquad\square$

