# OpenReview forum: "Towards certifying $\ell_\infty$ robustness using Neural networks with $\ell_\infty$-dist Neurons"
_ICLR.cc/2021/Conference — Reject_

### Official Review · AnonReviewer2 · 2020-10-15
**Review for new robust neural network definition**

**Rating:** 4
**Confidence:** 3

**Review:**

In this paper, the authors consider the task of adversarial/robust learning with respect to neural networks. The problem is a well-motivated one: suppose there is a neural network that on input a training set T={(x_i,y_i)} does a good classification job, but an adversary comes along and modifies some parts of T, then it is very possible that the neural network will classify almost incorrectly and hence isn't robust. In this paper they consider this setting and ask if we can naturally make neural networks robust.

In this direction, the authors propose a major change: in the conventional neural networks, one has sigmoid function which on input x, outputs sigma( w^T x ) (let's ignore bias for the time being) and the authors observe this functions is neither Lipschitz nor is it robust to noise. Instead, the author propost an ell_inf neuron which is just || w - x ||_inf. In this direction, they consider a neural network that is built out of ell_inf neurons. Simply by definition, it isn't too hard to see that this function is Lipschitz with respect to the ell_inf norm. The authors go on to show that every function can be represented using this new ell_inf NNs with sufficiently many neurons and sufficiently large depth. Furthermore, they go on to perform certain simulations for MNIST data.
In my opinion here are the pros and cons of the paper:

1) Pros: I think the problem is very well motivated and has been extremely well-studied. I am not an expert in this area, but i find their ell_inf neuron pretty interesting as well. Their simulations also seem very intriguing that such neurons seem to work after all (which is slightly surprising to me based on what I say next)

2) Cons: In my opinion, the authors do not make a sincere effort to compare both the models. A simple example where the new model is *extremely* inefficient is simply to compute the inner product function. It is easy to do it in the standard neural model (albeit its not robust), but in the new model, even the non-robust setting, I don't think the inner product can be computed easily. SO it seems to me that their "fix for robustness"might lack the decades of research that has been done in understanding and proving results about the standard sigmoid function. This is an important aspect which is missing in their work.

Overall, I think the idea is nice, but I'd tend towards rejection since their fix could be nice if they can show that everything computable in the standard NN model can be computed in their new ell_inf NN model (with approximately the same complexity), but this seems to be missing in its current form.

---

> ### Author Response · Authors · 2020-11-21
> **Response to AnonReviewer2**
>
> Thank you for your valuable feedback.
>
> In our opinion, the universal approximation theorem already guarantees arbitrary expressive power over the Lipschitz function class as long as the network size is big enough. Therefore, in general, our proposed L_{\infty} net can approximate any inner product network.
>
> As the two types of networks are different, it is true that L_{\infty} net with limited width and depth may be not easy to approximate the inner product; but please note that there is no ground truth that dot-product is the optimal design choice: in the experiment, both networks can learn good models in different tasks.
>
> We hope our response can address your concerns of the paper.

---

### Official Review · AnonReviewer4 · 2020-10-25
**Simple and effective proposal for $\ell_{\infty}$ certifiable robustness. Experiments could be more comprehensive.**

**Rating:** 6
**Confidence:** 4

**Review:**

**Summary:**

The contribution of the paper is threefold: First, it proposes a novel variation of AdderNet (Chen et al. 2020) that ensures the network is always 1-Lipschitz with respect to $\ell_{\infty}$ norm. The architecture allows one to generate robustness certificates with respect to $\ell_{\infty}$ norm with only a single forward pass, which is computationally cheap compared to many of the previous methods. Second, it analyzes the expressive power and robust generalization of the architecture. Finally, it proposes two training techniques (bias batchnorm and p-norm schedule) that overcome the optimization problem inherent with training $\ell_{\infty}$ Net and analyzes its empirical performance with respect to the previous methods.


**Strength:**
- The approach is easy to implement while performing reasonably well.
- Similar to (Anil et al. 2019), the approach is very efficient requiring only a single forward pass for certification, and it performs substantially better on MNIST in comparison with (Anil et al. 2019).
- The expressiveness of the architecture is analyzed, though the proof is simple, it is nice to settle the issue within the paper.
- I like that they are studying  $\ell_{\infty}$ robustness since it has been shown to be a more important problem compared to $\ell_{2}$ robustness (Goodfellow et al. 2018). This is in contrast with the recent popular randomized smoothing approach, which tends to focus more on $\ell_{2}$ robustness.


**Weakness:**
- The experiments do not include results for the convolutional variant of the model nor results for CIFAR-10. It is odd that the convolutional variant of the model is not included since the original AdderNet was developed specifically for convolution. Even though (Anil et al. 2019) did not include results for CIFAR-10, all the other related papers (Wong et al. 2018, Sven 2018, Zhang 2019)  included results for CIFAR-10. I don’t think performance on these experiments is a deal-breaker, but it is important to include these results so that future researchers can build upon your work and put your work in the context of existing literature.

**Recommendation**

Overall, I like the paper and recommend acceptance. I would further raise the score if the experimental concern above is addressed.

I find enforcing the 1-Lipschitz property through architecture to be a very promising direction for efficient robustness certification, and the work proposes an interesting and reasonably effective method for enforcing the 1-Lipschitz property with respect to $\ell_{\infty}$ norm. My main concern is that the experiments are not comprehensive enough. It is okay if it does not significantly outperform the previous methods, but it would be easier for the community to build upon your work if experiments are more comprehensive.


**Miscellaneous:**
- Since the approach is so similar to AdderNet, I recommend giving more credit to it. I wouldn’t have realized the similarities between them unless I dug into the citation.
- In Table 1 and 2, why are IBP, CROWN-IBP, and GroupSort considered as not scalable in the comparison? IBP & CROWN-IBP only cost ~2 times for certification compared to normal inference. Groupsort network requires only a single forward pass to calculate the certified radius.
- The generalization bound looks nice but I find that it slightly distracts from the main point of the paper. I am not too keen on the generalization bound that depends on Rademacher complexity in general since it does not consider the model learnable through gradient descent (not that it is an easy thing to do). This comment is more for future references, and I am okay with the way the paper is laid out right now.


[1] Chen, Hanting, et al. "AdderNet: Do we really need multiplications in deep learning?." Proceedings of the IEEE/CVF Conference on Computer Vision and Pattern Recognition. 2020.

[2] Goodfellow, Ian. “Defense against the Dark Arts: An overview of adversarial example security research and future research directions.” https://www.iangoodfellow.com/, 2018, https://www.iangoodfellow.com/slides/2018-05-24-DLS.pdf. Accessed 2020.

[3] Anil, Cem, James Lucas, and Roger Grosse. "Sorting out lipschitz function approximation." International Conference on Machine Learning. 2019.

[4]Wong, Eric, and Zico Kolter. "Provable defenses against adversarial examples via the convex outer adversarial polytope." International Conference on Machine Learning. PMLR, 2018.

[5]Gowal, Sven, et al. "On the effectiveness of interval bound propagation for training verifiably robust models." arXiv preprint arXiv:1810.12715 (2018).

[6]Zhang, Huan, et al. "Towards Stable and Efficient Training of Verifiably Robust Neural Networks." International Conference on Learning Representations. 2019.

---

> ### Author Response · Authors · 2020-11-21
> **Response to AnonReviewer4**
>
> Thank you for your valuable feedback.
>
> We recently implement convolutional neural networks and set all the neurons using l_\infty norm. The model architecture is almost the same to the large model in IBP and CROWN-IBP (5 convolutional layers and 2 fully connected layers; The widths of the convolutional layers are 64 or 128). The neural network is successfully trained on the MNIST dataset and Fashion-MNIST dataset. The performance of the l_\infty CNN models are similar to that of the MLP models in our paper. For example, the certified accuracy on MNIST and FashionMNIST dataset can still reach over 91% and 73% respectively. The result shows that our neuron can be well applied to convolutional networks.
>
> The main focus of this work is to provide an entirely novel neural network design with comprehensive theoretical supports. This model is different from all previous approaches in that it certifies robustness in nature. We provide experiments using MLP and CNN on MNIST and FashionMNIST to demonstrate our proposal. We would like to leave the study of more challenging datasets (e.g., CIFAR10) as future work.
>
> Regarding your detailed questions:
>
> Miscellaneous 1: Thanks for the question. In fact, our network does not require any multiplication, but AdderNet uses multiplication for the first convolution layer and the final linear layer. Standard BatchNorm is also used for all intermediate layers in AdderNet, which is shown to be essential. We use our proposed normalization (without the rescale operation) to retain the Lipschitzness of the model.
>
> Miscellaneous 2: For CROWN-IBP, the original paper claimed a computational complexity proportional to the number of classes, which can be large on CIFAR-100 or ImageNet. We notice that very recently, this issue seems to be overcome. For GroupSort, there is a projection operation in every layer that projects each row of weight matrix on the L_1 ball. This operation must sort the entire vector (as discussed in Appendix C in their paper), which is hundreds of times slower than the dot product and is prohibited if the network width is large. We will fix the scalability of IBP and CROWN-IBP in the next version.
>
> We hope our response can address your concerns of the paper.

---

### Official Review · AnonReviewer1 · 2020-10-27
**Interesting technique but needs more thorough evaluation**

**Rating:** 4
**Confidence:** 5

**Review:**

This paper proposes a new kind neural network based on a new kind of activation function, the L_\infty-Dist neuron, which they then demonstrate how to train, and show is both experimentally and certifiably robust.  Furthermore, they provide a theoretical result demonstrating that the network can approximate any desired function.

In terms of novelty, this paper has a lot of promise.  In particular:
* This approach to producing robust networks appears promising and fundamentally different from other approaches which either certify preexisting networks or smoothing approaches built on top of preexisting architectures.
* The theoretical result that their network can approximate lipschitz continuous functions is a helpful addition when considering an entirely new architecture.
* The training approach for a network without multiplications seems like a novel contribution in its own right.

However, the experimental section is lacking which diminishes my score.  Specifically, my main issue is that the utility of the method is dependent on its advantage over prior robust training and certification algorithms, and here it is only shown to perform similarly to prior approaches on the easy datasets of MNIST and FashionMNIST.  To properly evaluate the method, more difficult datasets need to be demonstrated, in particular - CIFAR10.   On Fashion-MNIST, no comparison is made to IBP in standard accuracy and while it outperforms CAP, cap is known to underperform other methods on other datasets.

=======================================================================
Update:

I thank the authors for providing additional data, however the additional data is insufficient for me to recommend acceptance.   While the approach is certainly novel, it appears to performs worse in the relevant metrics than other methods while working on less standard networks.  As the networks are so far from standard, it is necessary to see how they (and the method) behave on commonly accepted datasets.

Furthermore, AnonReviewer4 pointed out the similarities to AdderNet which I had overlooked.   Given these similarities I expect a more thorough methodological and experimental comparison to the original AdderNet.

---

> ### Author Response · Authors · 2020-11-21
> **Response to AnonReviewer1**
>
> Thank you for your valuable feedback. We recently implement convolutional neural networks and set all the neurons using l_\infty norm. The model architecture is almost the same to the large model in IBP and CROWN-IBP (5 convolutional layers and 2 fully connected layers; The widths of the convolutional layers are 64 or 128). The neural network is successfully trained on the MNIST dataset and Fashion-MNIST dataset. The performance of the l_\infty CNN models are similar to that of the MLP models in our paper. For example, the certified accuracy on MNIST and FashionMNIST dataset can still reach over 91% and 73% respectively. The result shows that our neuron can be well applied to convolutional networks.
>
> The main focus of this work is to provide an entirely novel neural network design with comprehensive theoretical supports. This model is different from all previous approaches in that it certifies robustness in nature. We provide experiments using MLP and CNN on MNIST and FashionMNIST to demonstrate our proposal. We would like to leave the study of more challenging datasets (e.g., CIFAR10) as future work.
>
> We have run IBP and CROWN-IBP based on the official github repo and perform a grid search over hyper-parameters. The best results are:
> IBP: standard acc 85.17; robust acc 81.33; certified acc 77.50;
> CROWN-IBP: standard acc 85.48; robust acc 81.34; certified acc 77.85;
> Our network has higher standard accuracy than both IBP and CROWN-IBP.
>
> We hope our response can address your concerns of the paper.

---

> > ### Comment · AnonReviewer1 · 2020-11-24
> > **At the expense of certified accuracy**
> >
> > To tabulate:
> >
> > MNIST:
> >
> > |                       |  St     |   Rb |  Cert  |
> > |------------------|--------|-------|--------|
> > |CROWN-IBP  | 98.18 | 93.95 |  92.98|
> > |Linfty-dist      | 98.61 | 93.78 | 91.59 |
> >
> > Fashion-MNIST:
> >
> > |                       |  St     |   Rb     | Cert |
> > |------------------|--------|---------|--------|
> > |CROWN-IBP  | 85.48| 81.34 | 77.85|
> > |Linfty-dist      | 87.46| 75.44 | 73.23|
> >
> > In 2 out of 3 metrics evaluated on the only two datasets you use, Linfty-dist method performs worse than CROWN-IBP.    In the first metric, standard accuracy, Linfty-dist performs slightly better than CROWN-IBP.  However, it is already known how to achieve high standard accuracy at the expense of robust accuracy or certified accuracy.  One can either train without a defense, or with PGD, or even use a provable training loss mixed with a standard loss.    In 3 out of the 6 total comparisons here, Linfty-dist in fact does significantly worse than the prior methods.    81.34% to 75.44% is nearly a 6% loss of robustness, and 77.85% to 73.23% is nearly a 5% loss of provable robustness, all for a 2% gain in standard accuracy, which was already achievable.     On MNIST a percentage of certifiable accuracy is lost for half a percentage gained in standard accuracy.
> >
> > Showing CIFAR10 here is absolutely necessary for me to recommend acceptance, as typically such gains and losses are magnified on this dataset, and it is not significantly more time consuming to run than Fashion-MNIST.
> >
> > It is also concerning that convolutions do not outperform MLP, but rather perform just as well.

---

### Official Review · AnonReviewer3 · 2020-10-28
**Review for Paper2097**

**Rating:** 5
**Confidence:** 5

**Review:**

This paper discussed a technique to obtain a neural network with verifiable
robustness guarantee. The approach is based on a redesign of neural network
building blocks. Instead of using matrix multiplication, the authors propose
to use the L_\infty norm operator, which is itself non-linear and Lipschitz.
The authors successfully train L_\infy network with several training tricks on
MNIST and Fashion-MNIST, and the verified accuracy is close to the
state-of-the-art results.

Strength:

1. The idea of using L_\infty operations as basic building blocks of a neural
network is simple. It provides verifiable robustness in a new way, and can be
useful for future works on improving verifiable robustness.

2. The authors prove that a Linf-dist net is an universal function
approximator for bounded 1-Lipschitz functions.

3. The authors discuss a few training tricks to train L_\infty net, such as
using an increasing p norm (from a small value to infinity) during training.

4. The paper is overall well written, well motivated and organized well.

Weakness:

1. Only very limited empirical evaluation is done on two small datasets.
Especially, it is claimed that the proposed approach is "scalable", yet only
results on small datasets are shown. Actually, in my opinion IBP is also
scalable; CROWN-IBP has recently been scaled up to ImageNet as well (according
to information on their github [1]).

2. It is unclear if L_\infty net can be extended to convolutional neurons to
work on larger datasets such as CIFAR. I think it should be possible to use
convolutional neurons with weight sharing here, so if the authors can include
convolutional L_\infty net results that will be a big plus.

3. The approach cannot outperform existing baselines (IBP and CROWN-IBP) on
MNIST and Fashion-MNIST. But I am okay with it. If the authors can show L_\infy
net on more challenging dataset such as CIFAR-10, it might be able to
outperform previous baselines.

Other points:

1. In table 2 why there is no standard acc and robust acc for IBP? And why not
include CROWN-IBP results as in Table 1 (it should be quite easy to change
MNIST to Fashion-MNIST in training).

2. In section 3.1 an inaccurate reference is given for the NP-completeness of
robustness verification. The correct paper to cite is Katz et al. [2], which
gives the proof in section I. The currently cited paper discussed complexity of
approximate verification algorithm which is a different setting.

3. Is it possible to generalize the L_\infty network to other general p norms
such as the L2 norm? A recent work on L2 norm verifiable training is [3].


Despite that the evaluation of the proposed algorithm is relatively weak, I
appreciate this approach and I am overall positive with this paper. I hope the
authors can provide more evaluation results as mentioned above during the
discussion period to make this paper stronger. I will consider increasing my
rating based on the author's response.


[1] Xu, Kaidi, et al. "Provable, Scalable and Automatic Perturbation Analysis on General Computational Graphs." NeurIPS 2020. https://arxiv.org/pdf/2002.12920

[2] Katz, Guy, et al. "Reluplex: An efficient SMT solver for verifying deep neural networks." International Conference on Computer Aided Verification. Springer, Cham, 2017.

[3] Singla, Sahil, and Soheil Feizi. "Second-Order Provable Defenses against Adversarial Attacks." ICML 2020. arXiv:2006.00731 (2020).

---

> ### Author Response · Authors · 2020-11-21
> **Response to AnonReviewer3**
>
> Thank you for your valuable feedback. We answer each of your concerns below.
>
> Regarding the term scalable:
>
> Thanks for the question. We claim our method is scalable since no matter how deep the network is, we can simply calculate the certified radius of any data point in a single forward pass without additional computation. Also, the network can be trained using the standard loss function but does not need adversarial training.
>
> Regarding using l_\infty net to convolutional neurons.
>
> We follow your suggestion to implement convolutional neural networks and set all the neurons using l_\infty norm. The model architecture is almost the same to the large model in IBP and CROWN-IBP (5 convolutional layers and 2 fully connected layers; The widths of the convolutional layers are 64 or 128). The neural network is successfully trained on the MNIST dataset and Fashion-MNIST dataset. The performance is similar to that of the MLP models in our paper. For example, the certified accuracy on MNIST and FashionMNIST dataset can still reach over 91% and 73% respectively. We will add these empirical results in the next version.
>
> Regarding CIFAR10.
>
> Thanks for the suggestion. The main focus of this work is to provide an entirely novel neural network design with comprehensive theoretical supports. This model is different from all previous approaches in that it certifies robustness in nature. We provide experiments using MLP and CNN on MNIST and FashionMNIST to demonstrate our proposal. We would like to leave the study of more challenging datasets (e.g., CIFAR10) as future work.
>
> Regarding 'Other point1'
>
> We have run IBP and CROWN-IBP based on the official github repo and perform a grid search over hyper-parameters. The best results are:
> IBP: standard acc 85.17; robust acc 81.33; certified acc 77.50;
> CROWN-IBP: standard acc 85.48; robust acc 81.34; certified acc 77.85;
> Our network has higher standard accuracy than both IBP and CROWN-IBP.
>
> Regarding 'Other point2'
>
> Thank you for pointing it out. We will fix it in the next version.
>
> Regarding 'Other point3'
>
> Our technique can be extended to any L_p norms with certified robustness guarantee in a straight-forward way. The universal approximation theorem still holds with some modifications.
>
> We hope our response can address your concerns of the paper.

---

> > ### Comment · AnonReviewer3 · 2020-11-23
> > **Thank you for the response.**
> >
> >
> > Thank you for the clarifications on my concerns. I appreciate the new results on training CNN on MNIST/Fashion-MNIST and the added IBP and CROWN-IBP baselines for Fashion-MNIST.
> >
> > Regarding scalability, I believe it can only be demonstrated by training using larger datasets such as CIFAR-10 and TinyImageNet. This paper looks very promising and I like it's idea, but as also pointed out by other reviewers, its empirical evaluation is insufficient and cannot really show the benefits of the proposed algorithm. So unfortunately I cannot increase my rating.
> >
> > Additionally, it seems the paper draft has not been revised yet. I hope the authors can take the opportunity during discussion period to revise their paper and update a new revision with new results. Especially, more discussions on CNN implementation and training are necessary. Are L_inf CNN networks easier or harder to train compared to MLPs? Why it has almost no improvement compared to MLPs (for other methods CNNs are usually better than MLPs)?

---

### Decision · Program_Chairs · 2021-01-07
**Final Decision**

**Decision:**

Reject

**Comment:**

In this paper, the authors propose a theoretically principled neural network that inherently resists ℓ∞ perturbations without the help of adversarial training. Although the authors insist to focus on the novel design with comprehensive theoretical supports, the reviewers still concern the insufficient empirical evaluations despite the novel idea and theoretical analysis.